# AI Red Teaming Through the Lens of Measurement Theory

**Alexandra Chouldechova**[*]
Microsoft Research

**A. Feder Cooper**
Microsoft Research

**Abhinav Palia**
Microsoft

**Dan Vann**
Microsoft Research

**Chad Atalla**
Microsoft Research

**Hannah Washington**
Microsoft

**Emily Sheng**
Microsoft Research

**Hanna Wallach**
Microsoft Research

## Abstract

Red teaming—i.e., simulating attacks on computer systems to identify vulnerabilities and improve defenses—can yield both qualitative and quantitative information about generative AI (GenAI) system behaviors to inform system evaluations. This is a very broad mandate, which has led to critiques that red teaming is both everything and nothing. We believe there is a more fundamental problem: various forms of red teaming are more commonly being used to produce quantitative information that is used to *compare* GenAI systems. This raises the question: (When) can the types of quantitative information that red-teaming activities produce actually be used to make meaningful comparisons of systems? To answer this question, we draw on ideas from measurement theory as developed in the quantitative social sciences, which offers a conceptual framework for understanding the conditions under which the numerical values resulting from a quantification of the properties of a system can be meaningfully compared. Through this lens, we explain why red-teaming attack success rate (ASR) metrics generally *should not* be compared across time, settings, or systems.

## 1 Introduction

Evaluating an AI system[2] requires making value judgments. Is the system good enough (for its intended purposes)? Is it better or worse than it was before an algorithm update? Is it better or worse than another system (designed for the same intended purposes)? Making such value judgments necessarily entails comparisons: comparing something about the system's behaviors (and associated capabilities or risks) to a particular value or threshold, to its own behaviors at an earlier point in time, or to another system's behaviors. Of course, we cannot make such comparisons without information about that system's behaviors and, in some cases, those of other systems. Such information can be either qualitative, as in the case of interviews with users about their experiences using the system, or quantitative, as in the case of computing accuracy rates for answering yes-no questions. In either case, though, such information must facilitate meaningful comparisons for it to useful for system evaluation [23].

Red teaming—i.e., simulating attacks on computer systems to identify vulnerabilities and improve system defenses [e.g., 11]—has emerged as a widely used approach for surfacing both qualitative and quantitative information about system behavior in evaluations of generative AI (GenAI) systems. The expanding variety of both red-teaming practices and the information they produce makes the exact role of red teaming in evaluation difficult to pin down. This slipperiness has been the focus of multiple stud-

---

[*]Corresponding author: `alexandrac@microsoft.com`

[2]We use the term "system" to refer either to a single model integrated with software and hardware that enables its use for inference, or to a hosted software service that incorporates one or more models to do the same.

NeurIPS 2024 Workshop on Safe Generative AI (SafeGenAI).

ies that have attempted to clarify the main purposes of red teaming [e.g., 8, 9, 14]. Despite this complexity, red-teaming procedures have tended to rely on a common metric for estimating and comparing system vulnerabilities: attack success rate (ASR). In this paper, we study ASR in order to try to clarify how red teaming fits into the GenAI evaluations landscape. Rather than trying to capture exactly what red teaming currently is or is not, we attempt to answer and draw lessons from a very specific question:

*Can ASRs be meaningfully compared across time, different systems, or different settings?*

To answer this question, we discuss, from first principles, what types of information red teaming would need to produce in order to facilitate meaningful comparisons of different systems. To ground our discussion, we briefly survey prior work in GenAI that explains how the expanding boundaries of red teaming make it difficult to understand what red teaming is (Section 2). We highlight that this issue, in turn, makes it difficult to understand what information can and cannot be obtained from red teaming. This analysis surfaces several key questions that must be answered to understand how red teaming could potentially play a role in making meaningful comparisons between systems, with a particular focus on comparing ASRs. To assist with identifying answers, we find it useful to turn to *measurement theory* from the social sciences (Section 3). Measurement theory can serve as a lens for critically examining red teaming, helping us disentangle when the information obtained from red teaming can facilitate meaningful comparisons and when it cannot (Sections 4).

## 2 Red teaming and its internal tensions

Red teaming—i.e., simulating attacks on computer systems to identify vulnerabilities and improve system defenses[3]—comes from the world of cybersecurity [5, 9, 19, 22]. The AI community has adopted red teaming as a key stage of GenAI system evaluation. In doing so, the community has expanded what constitutes red teaming to encompass a wide range of practices for probing GenAI systems for a wide range of issues [8, 9, 14], including security vulnerabilities, privacy leaks, memorization of intellectual property, stereotyping of marginalized groups, hate speech, and so on [12, 15, 17, 18, 20]. For brevity, we refer to all of these properties generically as "undesirable behavior." In this section, we discuss the ever-expanding surface area of red-teaming approaches and their fundamental limitations. We introduce attack success rate (ASR), which exemplifies these challenges and which will be our running example in the sections that follow for discussing numerical comparisons across systems.

**Ever-expanding surface area.** Red teaming practices vary along a number of dimensions. In the context of cybersecurity, red teaming is typically performed manually by teams of humans [2, 9]. However, in the context of GenAI, semi- or fully-automated approaches for red teaming are increasingly common [4, 24, 25]. GenAI systems are now routinely used both to generate attack inputs and to automatically determine whether an attack was successful by assessing system outputs using established success criteria [4, 6, 20].

Separate from the human versus automated red-teaming axis, there are several other dimensions along which red-teaming practices can vary, which we summarize from existing surveys: (1) expertise of participants in a red-teaming exercise (e.g., are red teamers domain experts, crowd-source workers, undergraduate students, etc?); (2) diversity of participants (e.g., which countries are red teamers from?); (3) system modality (e.g., text-based chatbots, text-to-image systems); and (4) scope of the red-teaming exercise (e.g., open-ended tasks or focused on specific undesirable behaviors) [9, 14, 19]. These dimensions suggest an enormous design space for possible red-teaming exercises. And this design space continues to expand: the range of each of these dimensions continues to grow over time, and altogether new dimensions get introduced into the mix. In all, there remains what the Frontier Model Forum has described as "a lack of clarity on how to define 'AI red teaming' and what approaches are considered part of the expanded role it plays in the AI development life cycle" [22].

**Fundamental limitations.** While on the one hand, red teaming is critiqued for having an enormous and expanding mandate, on the other, many have simultaneously argued that red teaming is inherently limited. For instance, Friedler et al. [10] combine observations about the Generative AI Red Team (GRT) challenge at DEFCON 2023 with a review of the literature to argue that red teaming "cannot

---

[3]One of the earliest usages of the term in a GenAI context refers to red teaming as "continuously interrogate the model's capabilities for possible problems (e.g., bias, misuse, safety concerns, etc.)" [21, p. 2].

effectively assess and mitigate the harms that arise when [AI] is deployed in societal, human settings." Frontier model developers like Anthropic have similarly argued that red teaming has gaps; Anthropic has characterized red teaming as a "qualitative approach" that "can serve as a precursor to building automated, quantitative evaluation methods" [3], rather than an approach that constitutes such evaluations on its own. This presents a contradiction: red-teaming practices and their adoption in new contexts both continue to grow, and yet this growth is occurring without fully attending to known limitations or uncovering unknown limitations of existing uses. This contradiction muddles understandings about what can be learned from red teaming; it is partly why there are often conflicting views on what red teaming is (and is not), and when it does (or does not) work. Without first resolving this contradiction, we cannot reasonably understand what can be gleaned from red-teaming practices and their their connections to other approaches to evaluating GenAI for undesirable behavior.

**Attack success rates as metrics for GenAI system comparison.** The slippage between red teaming and other forms of evaluation comes into stark relief when one examines how attack success rates (ASR) from red teaming activities get (mis)used in making *comparative claims* about GenAI systems. It is fairly common in red-teaming research papers to have some (automated or manual) binary classifier determine if a particular attack was successful at inducing a system to exhibit an undesirable behavior [e.g. 12, 16, 24]. These attack success rates are often used as *metrics*: aggregating attack-success bits into per-system averages over attacks, and comparing these averages across different systems to judge overall and relative robustness. For example, HarmBench [16] develops a suite of attacks, computes average ASR across open-source models, and then directly compares these averages. At face value, as an average of bits, it may seem like ASR is relatively straightforward to compute. However, the details of defining attack success—of determining the values of particular bits—requires great care. As we discuss in more detail in Section 4, without exercising great care, ASR metrics cannot be meaningfully compared across time, different systems, or different settings.

## 3   Using measurement theory for valid measurement

When can attack success rate (ASR) metrics be meaningfully compared? To answer this question we must first more clearly understand what is being asked. Consider a concrete hypothetical example in which a group engages in manual red-teaming activities conducted on two systems, $L_1$ and $L_2$. The team obtains an ASR of $a_1 = 0.1$ for $L_1$ and an ASR of $a_2 = 0.2$ for $L_2$. What can we say about the systems $L_1$ and $L_2$ on the basis of these ASRs? What comparative statements are we justified in making? There is no doubt that, as numbers, $a_2 = 0.2 > 0.1 = a_1$: the ASR reported for system $L_2$ is higher than that reported for system $L_1$. This is a *descriptive* statement summarizing the outcomes of the specific sets of attacks, observed system outputs, and determinations of success. But can we conclude from these results that system $L_2$ is *more vulnerable* than $L_1$?

This question is akin to, but (in certain ways) more complex than, familiar questions arising in observational and experimental studies. Consider a hypothetical randomized controlled trial where upon discharge patients are randomized to either the current standard of care or a new treatment. For the control group, we observe a 30-day hospital readmission rate of $a_2 = 0.2$, compared to $a_1 = 0.1$ for patients in the treatment group. Is the treatment effective at reducing hospital re-admission? Once again, there is no doubt that, as numbers, $a_2 = 0.2 > 0.1 = a_1$: the control group patients in our study were twice as likely to be re-admitted as treatment group patients. This is a descriptive statement about the observed outcomes among study participants. When asking about treatment effectiveness, however, we are not asking about the study sample. We are instead asking an *inferential* question about whether the expected readmission rate under treatment in the *broader population*, $\alpha_1$, is lower than the expected readmission rate under standard of care $\alpha_2$. The observed rates, $a_2$ and $a_1$ are *estimates* of these population parameters. At minimum, we would want to conduct a hypothesis test and/or report a confidence interval to determine whether we have sufficient evidence to conclude that $\alpha_2 > \alpha_1$.

Just as the question of treatment effectiveness is an inferential question—asking whether the study provides sufficient evidence to draw conclusions beyond the study sample itself—the question of comparing ASRs from red-teaming activities also has an inferential flavour. But meaningful comparison relies on more than simply accounting for *sampling variation* between estimates, $a_j$, of population parameters, $\alpha_j$. For a more complete picture, we turn to prior work from measurement theory in the quantitative social sciences.

**Measurement theory.** *Measurement* refers to the systematic quantification of properties of entities or events, resulting in numerical values—i.e., measurements—that can be meaningfully *compared*. Specifically, to be able to compare measurements, we require a high degree of *measurement validity*, which is the the extent to which a a measurement procedure measures what it purports to measure. The remainder of this section provides an overview of measurement theory and measurement validity.

Measurement approaches lie along a spectrum between *representational measurement* and *pragmatic measurement*. Representational measurement refers to expressing objects and their relationships using numbers. For example, metrology, the study of measurement in the physical sciences, is largely representational, giving rise to the familiar units such as length, mass, and time, that underpin scientific inquiry. Pragmatic measurement focuses on measuring abstract concepts that are not amenable to direct observation in ways that yield measurements with the "right sort of properties for [the] intended use" [13]. Pragmatic measurement most commonly arises in the social sciences, where many quantities of interest reflect concepts that are abstract, complex, and sometimes contested. Examples include the Gross Domestic Product (GDP) constructed to reflect the health of a country's economy, the Apgar score used to evaluate the condition of a newborn immediately after birth, and the various measures of democracy used in political science.

Many of the measurement tasks entailed in evaluating GenAI systems can be viewed as exercises in pragmatic rather than representational measurement. This is because the concepts to be measured— regardless of whether they are concepts related to a system's capabilities, like intelligence and reasoning, or concepts related to a system's risks, like stereotyping and anthropomorphism—are abstract, complex, and sometimes contested, much like the health of a country's economy, the condition of a newborn immediately after birth, and the democracy of a nation [23].

**Quantification is not the same as valid measurement.** The primary purpose of measurement is to facilitate meaningful comparisons: for example, to compare something (e.g., the temperature of the earth, the GDP) to itself over time; to compare something (e.g., the height of a child, a newborn's Apgar score) to a particular numerical value; and to compare multiple things (e.g., the weights of three bicycles, two nations' democracy scores) to one another.

Clearly, not all numbers produced by examining an entity can or should be meaningfully compared. For instance, the 3rd digit of of a bike's serial number is a *number*, but does not reflect a property that yields insight if compared across bikes. As a less trivial example, consider administering the Stanford-Binet intelligence test (SB5) to two 5 year-old children, Jane who scores 117 and Ruslan who scores 82. The SB5 has been developed and validated as a test of certain cognitive abilities, and is a prominent example of a measurement instrument in the psychometric tradition. Even so, if we then learn that the test was administered in English, a language that Jane speaks but Ruslan does not, it would still not be meaningful to compare the children's scores. Ensuring that the procedures adopted produce quantities that can be meaningfully compared—i.e., ensuring that we are doing *valid measurement*—is the primary challenge.

To assess measurement procedures for validity and to develop more reliable and valid procedures, social scientists rely on *measurement theory*. Measurement theory provides a conceptual framework for systematically moving from a concept of interest to measurements of that concept [1]. Measurement theory also provides a set of lenses for interrogating the reliability and validity of measurement instruments and their resulting measurements, including test–retest reliability, inter-rater reliability, face validity, content validity, convergent validity, discriminant validity, predictive validity, hypothesis validity, and consequential validity. Each lens constitutes a different source of evidence; together the evidence collected using these lenses can paint a comprehensive picture of reliability and validity.

**Measurement framework.** The core measurement framework applied within the social sciences involves four levels: the *background concept* or "broad constellation of meanings and under standings associated with [the] concept;" the *systematized concept* or "specific formulation of the concept, [which] commonly involves an explicit definition;" the measurement instrument(s) used to produce instance-level measurements; and, finally, the instance-level measurements [1]. As shown in the "Concept" column of the measurement framework shown in Figure 1, these levels are connected via three processes: *systematization*, *operationalization*, and *application*. For example, when measuring the prevalence of hate speech in a conversational search system deployed in the UK, the background concept might be a set of high-level definitions of hate speech like those provided above; the systematized concept might be a set of linguistic patterns that enumerate the various ways that speech acts

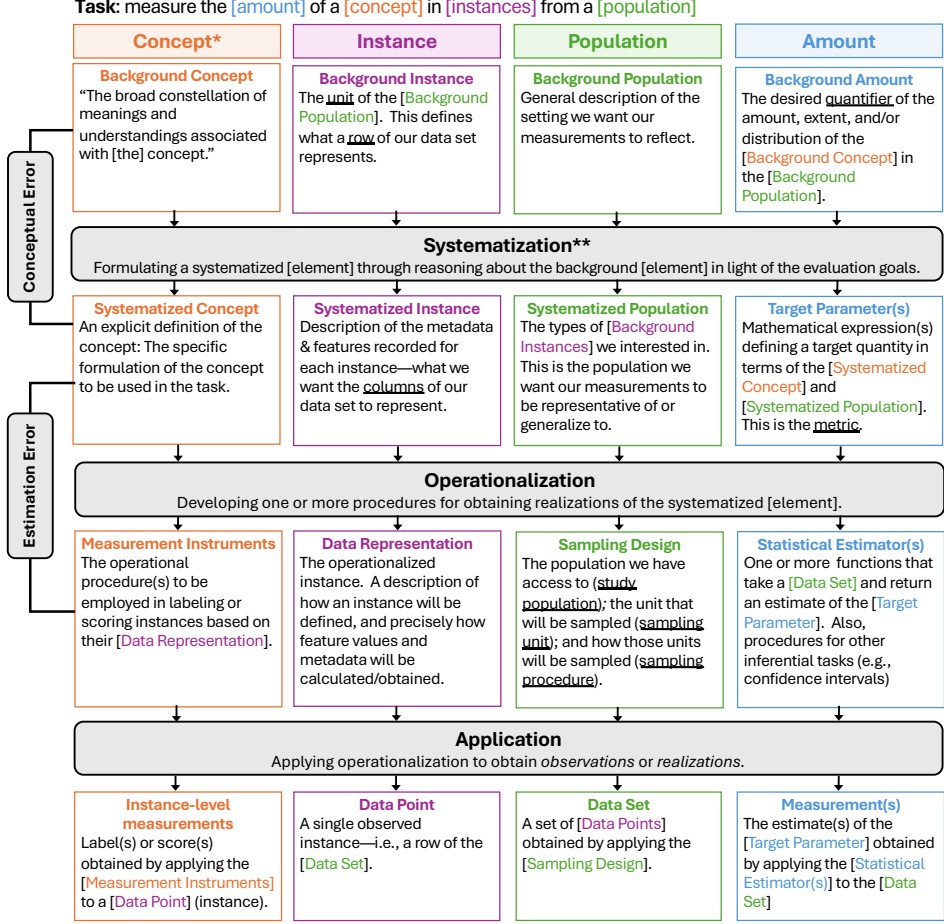

Figure 1: Measurement framework developed for tasks of the form: *measure the [amount] of a [concept] in a [population] of [instances].* The Concept column is adapted from prior work in political science [1]. The Instance, Population, and Amount elements are new. We use the term *Context* to refer to the Instance and Population elements. The table is reproduced with modification from related work by the authors [7].

can promote violence; the measurement instrument might be an LLM fine-tuned to identify those linguistic patterns; and the instance-level measurements might be a set of counts indicating number of linguistic patterns found in each conversation from a data set of conversations in the system's real-world deployment context. *Validity issues* arise as systematic gaps across the four levels, such as using a US-centric systematization of hate speech when evaluating a system in the UK context.

**Applying the same measurement procedure is not necessary or sufficient for valid measurement.** Applying different measurement procedures, or applying the same procedure in different settings, also does not invalidate our measurements. To understand why, consider two illustrative examples:

**Sufficiency** Repeating the same procedure in different settings is *not sufficient* for concluding that we have obtained valid measurements of the concept in both cases. Suppose we obtain a performance metric by running our system on a "validated" static benchmark. Several months later, a new system version is released, and we run the same benchmark and calculate performance. Unbeknownst to us, the system-development team included the benchmark corpus in the data used to train the latest system. This has *invalidated* the measurement instrument (benchmark), making it unusable for comparing the two system versions.

**Necessity** Repeating the same procedure is also *not necessary* for valid measurement and comparison. For instance, we may want to evaluate stereotype risk for a system deployed in Canada, which has two official languages, English and French. While we require a common *systematization* of stereotyping, we need different *operationalizations* (measurement instruments) for the two different languages and cultural contexts. The systematization needs to be *broad enough* to cover hate speech as it presents in both contexts. The measurement instruments, however, need to be suitably *tailored* to each target setting—at minimum, they need to use different languages.

Figure 1 introduces three other elements that, like the *concept*, need to be systematized, operationalized, and applied when performing measurement tasks: instance- and population-level *context* and the *amount*. This framework, which we developed in the process of evaluating GenAI systems [7], structures the process of developing and validating measurement tasks of the form: *measure the [amount] of a [concept] in a [population] of [instances].* For instance, we can frame the hate speech example as measuring the prevalence (amount) of hate speech (concept) in a conversational search system's responses (instances) in the current UK deployment (population). Alternatively, if we are interested in specifically adversarial interactions, we could specify the target population as "adversarial probing of the system in the current UK deployment." Whatever the task specification is, the framework helps us assess the validity of our resulting measurements—to determine whether they are indeed numbers that can be meaningfully compared—by clearly specifying each of the four levels for each element, and thus helping to surface gaps between them.

We devote remainder of the paper to examining red teaming through the lens of measurement theory. We focus specifically on determining when the information obtained from red teaming can facilitate meaningful comparisons across ASRs—and thus be used to make value judgments about GenAI systems—and when it cannot.

## 4   Red teaming through the lens of measurement theory

We now demonstrate how measurement theory can help us understand red-teaming practices in terms of entailed concept(s), context(s), and metric(s), with respect to ASR. Rather than inventorying the considerations relevant to planning or documenting red-teaming activities [9, 10, 14], we structure the task of understanding what (if anything) a given red-teaming activity is measuring.

We briefly provide two framing notes for the discussion that follows. First, we are using a very specific notion of measurement, as articulated in Section 3. The ability to do valid measurement—to compare metrics across systems or settings—comes not from *running the same procedure*, but from ensuring that we are targeting the same *systematized concept* using the same *quantifier* (metric) through *appropriately tailored* measurement instruments (operationalizations). Second, we emphasize that not all red-teaming activities can or should be interpreted as measurement—much less used as such. For example, red-teaming activities can function as "existence proofs," producing examples of undesirable behavior [e.g., 6, 17]. This section is therefore concerned with running attacks that culminate in quantities that one might try to—justifiably or not—compare across time, systems, or settings. To ground our discussion, Figure 2 provides an instantiation of the four column measurement framework in the setting of a generic single-turn attack red teaming activity. For clarity in presentation, we proceed in a different order from the columns in Figure 2, beginning with population.

**Population: What is an "attack", and what kinds of attacks are we interested in?** Red-teaming activities also vary widely in the types of attacks they use, thereby involving different distributions over possible *input* prompts and, as a result, producing quantities that are reflective of different population-level "contexts." In other words, these activities vary in their underlying *threat models*. We can show this with a very simple example. Consider two activities that differ only in their number of prompts allowed per attack, 10 or 1000. These are arguably two different threat models. Even if two attacks are identical for the first 10 prompts, the latter has 990 more turns to "succeed" where the former has failed. (For more on attack "success," see Concept, below.) As a result, these attacks involve different distributions over inputs and produce results for different populations.

Some activities focus on red-teaming exercises whose population-level contexts are more reflective of expected user behavior, whereas others focus on adversarial attacks. For manual or semi-automated red teaming, team composition has a large impact [2]. For example, diverse teams will generally have greater coverage of the input space; subject-matter experts will probe undesirable behavior in different ways than laypeople.

**Instance: How is the target system *output* generated during a red-teaming activity?** This is another important component when thinking about the evaluation context—specifically, the instance-level context. One such consideration is the number of tokens generated by the target system for a single attack. "Success" (discussed below) can hinge on such decisions. For example, Mazeika et al. [16] demonstrate how attack success rate drastically decreases as they increase the number of tokens generated.

**Concept: What does it mean for an attack to "succeed"?** Most red-teaming activities aim to identify undesirable behavior. We can view "undesirable behavior" as the common *background concept* for defining attack success. Activities range widely in what *systematized concepts*—i.e., different, precisely defined behaviors—are being targeted. Many activities fail to precisely specify an attack target altogether; they lack a clear threat model.

In automated red teaming, this presents as a leap straight from a background concept to an operationalization, i.e., a classifier for determining attack success—skipping the critical systematization step altogether. This can be especially problematic when the attack target falls under dissentive risk (i.e., "outcomes [that] are complex or contested" [9]), such as stereotyping or hate speech. This is because systematization plays an important role in resolving the types of definitional conflicts that are endemic to dissentive risk. Without systematization, these conflicts remain unresolved, which can lead to unclear or incorrect operationalizations. Even for consentive risks (i.e., where attack success conditions are clearly defined), such as divulging of personally identifiable information (PII), where one might argue there is a "natural" broadly understood systematization, activities often vary in their operationalizations.

**Amount: How is the ASR metric calculated?** The primary metric reported in red teaming is the *attack success rate* (ASR). ASR is defined as the proportion of attacks that elicit undesirable behavior from the target system. In automated red teaming, where success is assessed with a classification model rather than through manual inspection, it is common to describe different classifiers as giving rise to different ASR metrics (e.g., see the discussion of robust metrics in Mazeika et al. [16]). Within the measurement theory framework (Section 3), different classifiers arise as different operationalizations of the focal concept.

Having provided some analysis of red teaming through the lens of measurement theory, we can now revisit the question posed at the beginning of the paper.

**Can attack success rates be meaningfully compared across time, different systems, or different settings?** *In general, **NO**. By the time the conditions are satisfied to answer **YES**, we are arguably no longer performing "red teaming," as it tends to be conducted in the literature, but rather are engaged in a formal measurement exercise.*

Consider the simple case represented in Figure 2 of single-turn attacks where we get to observe a system *output* $Y = L(X)$ generated by system $L$ in response to an input prompt, $X$. Let

$$s(Y; X) = I(Y \text{ is undesirable}; X),$$

be the indicator that the attack has succeeded. Here $s(Y; X)$ is an "oracle" function that encodes the *systematized concept* criteria for determining whether observing system output $Y$ in response to input $X$ constitutes undesirable behavior. Given a threat model that specifies the distribution of possible attacks (inputs), $X \sim \mathcal{D}$, we can define the *target attack success rate metric* as,

$$\alpha(s, L, \mathcal{D}) := \mathbb{P}_{X \sim \mathcal{D}}(s(L(X); X) = 1). \tag{1}$$

This quantity depends on three things: the system, $L$, being evaluated; the attack success criterion, $s$; and the input distribution, $\mathcal{D}$. In this setup, we can view the *observed* attack success rate from a given red-teaming activity (represented in the Statistical Estimator(s) and Observed ASR cells of Figure 2) as *estimates* of $\alpha(s, L, \mathcal{D})$.

Now suppose we are interested in evaluating whether a new system, $L_2$, has a lower rate of undesirable behavior than system $L_1$. As discussed in the previous section, red-teaming activities differ widely in how they specify the success criterion, $s$, how they operationalize this criterion, $\tilde{s}$, and in the input distribution, $\mathcal{D}$. If either the success criteria or attack input distribution (threat model) is different between the red-teaming activities used for the two systems (i.e., $s_1 \neq s_2$ or $\mathcal{D}_1 \neq \mathcal{D}_2$), then comparing observed attack success rates ASR tells us little about which system is more prone to undesirable behavior. This is because comparing even the "true" target attack success rate metrics,

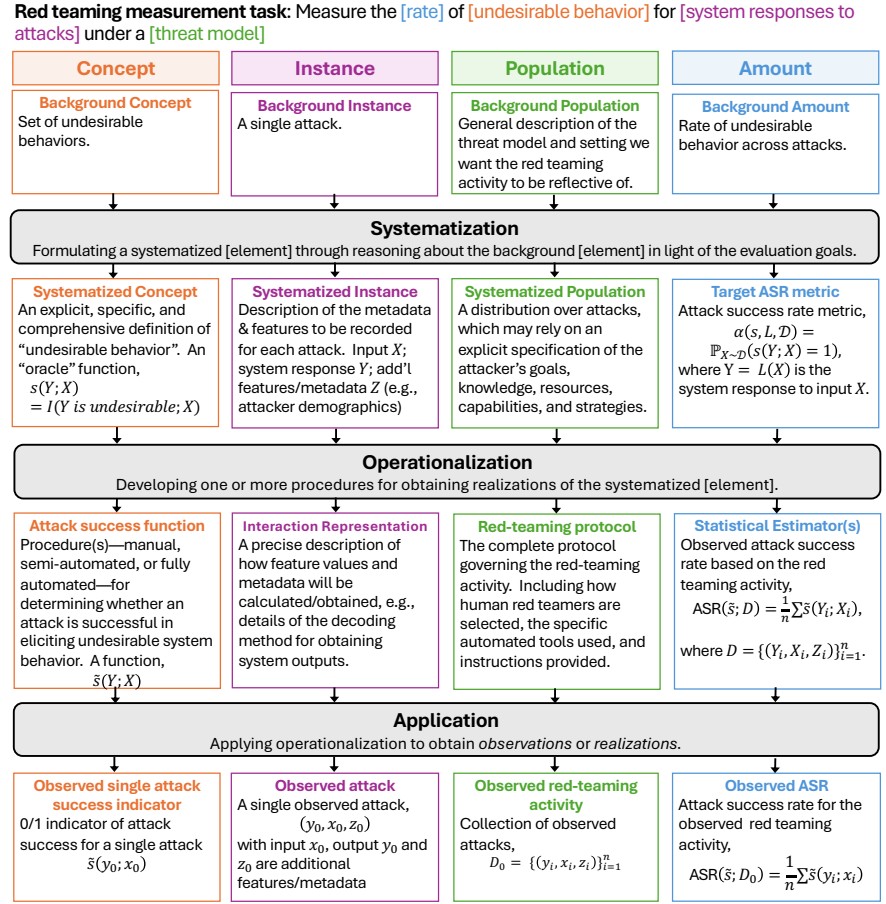

Figure 2: Measurement framework instantiated for a generic *single-turn* red teaming activity as a measurement task. We view the observed attack success rate, $\mathsf{ASR}(\tilde{s}, D_0)$, as an estimate of a *target* attack success rate metric, $\alpha(s, L, \mathcal{D})$, that is defined wrt a distribution over attacks.

$\alpha(s_1, L_1, \mathcal{D}_1)$ and $\alpha(s_2, L_2, \mathcal{D}_2)$, would not provide meaningful insight into which system is more prone to undesirable behavior. The observed ASRs are, at best, good estimates of these target $\alpha$'s, and are thus no more meaningful to compare.

But what if we have standardized our red-teaming activity using an automated approach to ensure that attacks come from the same distribution, $\mathcal{D}_0$ and that their success is assessed using the same criterion, $s_0$? In this case, comparing $\alpha(s_0, L_1, \mathcal{D}_0)$ to $\alpha(s_0, L_1, \mathcal{D}_0)$ *could* provide meaningful insight into undesirable behavior. But we could still have validity issues to contend with, for example, if the red-teaming protocol relied not on a fresh set of attacks drawn from $\mathcal{D}_0$ but instead reused a previous set of inputs that been used in training or putting safeguards on $L_2$, but not $L_1$, as discussed in Section 3.

## 5 Conclusion

Red teaming has grown to encompass a broad range of practices for probing GenAI systems for a wide range of vulnerabilities and undesirable behavior. This broad mandate has led to critiques that it is both everything and nothing. In this paper, we focused on the question: (When) can the types of quantitative information that red-teaming activities produce (in particular, ASRs) actually be used to make meaningful comparisons of systems? We drew on ideas from measurement theory from the social sciences to answer this question, exploring the the conditions under which the numerical values resulting from a quantification of the properties of a system can be meaningfully compared and explaining why red-teaming ASR metrics generally should not be compared across time, settings, or systems. Having articulated the scope of this problem, in future versions of this work we will articulate procedures for more meaningful comparisons of ASRs across time, different systems, and different settings.

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
