# OpenReview forum: "AI Red Teaming through the Lens of Measurement Theory"
_NeurIPS.cc/2024/Workshop/SafeGenAi — SafeGenAi Poster_

### Official Review · Reviewer_g3rr · 2024-10-09
**Review for paper titled "Red Teaming: Everything Everywhere All at Once"**

**Rating:** 5
**Confidence:** 3

**Review:**

Novelty: The core novelty lies in applying measurement theory from the social sciences to dissect when and how red teaming metrics—like Attack Success Rate (ASR)—can provide valid insights.
Reasons to accept: This paper provides a structured framework for understanding its limitations. The argument that ASR and other metrics from red teaming cannot always be meaningfully compared between systems or over time is supported by some illustrative examples
Reasons to reject:  This paper could benefit from empirical case studies or experiments comparing red teaming metrics across different AI systems and showing when they fail to provide meaningful insights.

---

### Official Review · Reviewer_eork · 2024-10-09
**application of measurement theory to ML evaluation is important but too theoretical**

**Rating:** 6
**Confidence:** 3

**Review:**

This paper tries to call attention to the issue in machine learning that attention to detail in experiment design and evaluation measurement is something that can be lacking in current scientific work, and industry benchmarks as well.

Unfortunately, the abstract focuses only on one aspect which is comparison of numerical attacks success rates. The paper is broader than this, and I think it could either narrow its focus or the abstract could try to explain more aspects. The title is worse, because it doesn't even hint that the paper will be talking about measurement or comparison. I think the title should hint at experiment design, measurement theory, metric design, and/or metric comparison, which seem to be the central themes of the paper.

The paper clarifies many terms from measurement theory that can be applied to better understand the role that red teaming plays in science. Section 2 is a great overview of red teaming that helps the reader understand the evolution of the term. The measurement theory is quite interesting but theoretical. I think the paper would be stronger with more examples of how to perform comparisons of LLM systems, concrete examples of less problematic metrics, suggestions for how to transform existing evaluation frameworks like HarmBench, etc.

Overall, I think this paper has important contributions and conveys them in a well-written manner. I think it falls short in suggesting practical metrics and approaches so that current researchers performing red teaming could have more takeaways. Since the theory is well established here, I think the biggest problem is communicating it effectively to a machine learning audience.

Pros
- identification of important problem lack of attention paid to evaluation metrics

Con
- abstract does not summarize full contributions of paper
- title does not suggest what will be present in the paper
- paper falls short in suggesting practical metrics and approaches for current researchers
- communicating effectively to practitioners is the big problem since measurement theory is known

Nits
- citations on lines 35, 61, 63, 69 are strangely formatted, the e.g. should be outside the brackets